# Molecular Characterization of Galectin-3 in Large Yellow Croaker *Larimichthys crocea* Functioning in Antibacterial Activity

**DOI:** 10.3390/ijms241411539

**Published:** 2023-07-16

**Authors:** Yao Yang, Baolan Wu, Wanbo Li, Fang Han

**Affiliations:** Key Laboratory of Healthy Mariculture for the East China Sea, Minsistry of Agriculture and Rural Affairs Fujian Provincial Key Laboratory of Marine Fishery Resources and Eco-Environment, Jimei University, Xiamen 361021, China; yangyaoyj@163.com (Y.Y.); wbl5160691422@163.com (B.W.); li.wanbo@jmu.edu.cn (W.L.)

**Keywords:** Galectin-3, agglutination, antibacterial activity, innate immunity, *Larimichthys crocea*

## Abstract

Galectins are proteins that play a crucial role in the innate immune response against pathogenic microorganisms. Previous studies have suggested that Galectin-3 could be a candidate gene for antibacterial immunity in the large yellow croaker *Larimichthys crocea*. In this study, we cloned the Galectin-3 gene in the large yellow croaker, and named it LcGal-3. The deduced amino acid sequence of LcGal-3 contains a carbohydrate recognition domain with two conserved β-galactoside binding motifs. Quantitative reverse transcription PCR (qRT-PCR) analysis revealed that LcGal-3 was expressed in all the organs/tissues that were tested, with the highest expression level in the gill. In *Larimichthys crocea* kidney cell lines, LcGal-3 protein was distributed in both the cytoplasm and nucleus. Moreover, we found that the expression of LcGal-3 was significantly upregulated upon infection with *Pseudomonas plecoglossicida*, as demonstrated by qRT-PCR analyses. We also purified the LcGal-3 protein that was expressed in prokaryotes, and found that it has the ability to agglutinate large yellow croaker red blood cells in a Ca^2+^-independent manner. The agglutination activity of LcGal-3 was inhibited by lipopolysaccharides (LPS) in a concentration-dependent manner, as shown in the sugar inhibition test. Additionally, LcGal-3 exhibited agglutination and antibacterial activities against three Gram-negative bacteria, including *P. plecoglossicida*, *Vibrio parahaemolyticus*, and *Vibrio harveyi*. Furthermore, we studied the agglutination mechanism of the LcGal-3 protein using blood coagulation tests with LcGal-3 deletion and point mutation proteins. Our results indicate that LcGal-3 protein plays a critical role in the innate immunity of the large yellow croaker, providing a basis for further studies on the immune mechanism and disease-resistant breeding in *L. crocea* and other marine fish.

## 1. Introduction

China, as a major aquaculture country, accounts for 60% of the world’s total aquaculture production [1]. The large yellow croaker (*Larimichthys crocea*) belongs to the order of Osteicthys, Perciforme, Sciaenidae, and is a warm–temperate offshore pelagic migratory fish [2]. It is the largest economically important marine fish species in China’s aquaculture industry. In 2022, the production of this fish was approximately 254,000 tons, making the large yellow croaker aquaculture one of the pillar industries in China’s economy. One of the main diseases affecting large yellow croaker aquaculture is visceral white spot disease caused by *Pseudomonas plecoglossicida* infection [3,4,5]. The infection rate is from over 70% to 80%, resulting in an annual economic loss of CNY hundreds of millions. The disease mainly occurs at low temperatures, and the use of antibiotics can lead to drug residues and increased antibiotic resistance [4]. Currently, there are no safe and effective methods for prevention and control. In order to elucidate the genetic mechanisms of resistance to visceral white spot disease in the large yellow croaker and achieve disease-resistant molecular breeding, our laboratory has conducted GWAS analysis on different populations of large yellow croaker that are resistant or susceptible to the infection of *Pseudomonas plecoglossicida*. We have identified Galectin-3 in the large yellow croaker as one of the candidate genes for resistance to visceral white spot disease [6]. Therefore, we are analyzing its molecular characteristics and antibacterial activity, hoping to provide theoretical references for understanding the genetic mechanisms of disease resistance in the large yellow croaker and other fish species, and laying a foundation for disease-resistant molecular breeding.

Galectins, a type of lectin, possess broad-spectrum antibacterial, antiviral, anti-parasitic, and antifungal activities, in addition to immune-regulatory functions [7]. Galectins have a relatively high affinity to β-galactoside, characterized as conserved carbohydrate recognition domains (CRDs) that carry conserved β-galactoside binding motifs [8]. In mammals, fifteen members of the galectin family have been identified, and they are categorized into three subtypes based on their CRD organization: prototype, tandem-repeat type, and chimeric galectins. Among these subtypes, Galectin-3 (Gal-3) is the only chimeric type found in vertebrates, and contains a CRD linked to an unusually long N-terminal domain that exhibits lectin-independent activity [7].

Galectin-3 plays a crucial role in recognizing and aggregating pathogens, regulating cell adhesion, activating the complement system, promoting phagocytosis, and regulating innate immunity [9]. Recent studies have revealed that Galectin-3 also exhibits immunological functions, such as antibacterial, antiviral, and antiparasitic activities in aquatic animals, and is an essential component of the fish immune system [10]. For example, YdGal-3 derived from yellow drum (*Nibea albiflora*) can effectively destroy the cell walls of various bacteria such as *Pseudomonas plecoglossicida*, *Vibrio parahemolyticus*, *V*. *harveyi*, *and Aeromonas hydrophila*, thereby playing a crucial role in the innate immunity of yellow drum [11]. SmLgals3 from turbot (*Scophthalmus maximus L*.) has a strong binding capacity to LPS, peptidoglycan (PGN), and lipoteichoic acid (LTA) [12]. Similarly, PoGal-3 from Japanese flounder (*Paralichthys olivaceus*) tightly binds to LPS and PGN and exhibits significant binding ability to Gram-negative and Gram-positive bacteria [13]. Additionally, a Galectin-3 fragment isolated from Atlantic salmon mucus (*Salmo salar*) has been found to exhibit Ca^2+^-independent coagulation activity and can aggregate the Gram-negative pathogen *Moritella viscosa* [14]. Galectin-3 in grass carp (*Ctenophyngodon idella*) is significantly upregulated in the liver and spleen upon stimulation with grass carp reovirus (GCRV), LPS, and polyinosinic-polycytidylic acid (poly I: C), suggesting its involvement in the immune response of grass carp [15]. OnGal-3 from Nile tilapia (*Oreochromis niloticus*) exhibits strong binding and aggregation activity against *Streptococcus agalactiae* and *A*. *hydrophila* and can enhance the phagocytic function of monocytes/macrophages by inducing the upregulation of inflammatory factors such as IL-6, IL-10, IL-8, and MIF in monocytes/macrophages, thereby enhancing the immune function of the body [16,17]. Finally, Galectin-3 in zebrafish (*Danio rerio*) can bind to the glycoprotein of the infectious hematopoietic necrosis virus (IHNV) in vitro and inhibit its adhesion to the surface of fish epithelial cells [18].

In this study, we identified and characterized large yellow croaker Galectin-3 (LcGal-3), including its structural features, phylogenetic relationship, and expression profile in different tissues after infection with *P*. *plecoglossicida*. We also purified recombinant LcGal-3 fusion protein for use in blood coagulation, bacterial agglutination, and antibacterial testing. Furthermore, by performing blood coagulation tests using LcGal-3 mutant (deletion and point mutation) proteins, we demonstrated that specific residues play a critical role in coagulation. Our findings provide the first demonstration of a chimeric galectin in the large yellow croaker, and offer new insights into the important functions of galectins, particularly chimeric galectins, in innate immunity in teleosts.

## 2. Results

### 2.1. Sequence Characteristics and Bioinformatics Analysis of LcGal-3

The open reading frame (ORF) of *LcGal*-*3* was found to be 1155 bp long, encoding a protein of 384 amino acids (Figure 1). The calculated molecular mass of LcGal-3 was 38.17 kDa, and its theoretical pI was 5.01. Using the SignalIP-5.0 and TMHMM Server v.2.0, we predicted that LcGal-3 protein lacks both signal peptide and transmembrane regions. NetPhos 3.1 Server predicted fifteen phosphorylation sites in LcGal-3, including eight serine (Ser), six threonine (Thr), and two tyrosine (Tyr) sites (Figure 1A). The SMART protein domain prediction indicated that LcGal-3 contains a galectin CRD (Figure 1B) that is rich in proline, glycine, and alanine at the N-terminal. We used the SWISS-MODEL database to model the homology of the LcGal-3 protein based on the human Gal-3 template, and its tertiary structure is shown in Figure 1C, generated by VMD 1.9.2 beta 1.

ClustalOmega analysis showed that the LcGal-3 protein is highly conserved among mammals and fish and shares similarities with the Gal-3 proteins of *Nibea albiflora*, *Lates calcarifer*, and *Scophthalmus maximus*, with a homology of 95.56%, 92.59%, and 91.85%, respectively. (Figure 2A and Table 1). We observed two highly conserved sugar binding motifs, H-NPR and WG-EE-(Figure 2A). Furthermore, phylogenetic tree analysis demonstrated that the LcGal-3 protein is clustered with fish Gal-3, and closely related to the Gal-3 of *Nibea albiflora* (Figure 2B); notably, *Larimichthys crocea*, *Nibea albiflora*, *Siniperca chuatsi*, *Amphiprion ocellaris*, and *Perca flavescens*, which all belong to the Perciformes, are grouped into one branch, while mammals, birds, and amphibians form another cluster (see accession numbers of each species in Table 1). These results are consistent with the traditional evolutionary relationships.

### 2.2. Tissue Distribution of LcGal-3 and Subcellular Localization of LcGal-3

To confirm the immune function of LcGal-3 in *L*. *crocea*, it is necessary to study its tissue expression and regulation after pathogen stimulation. We used qRT-PCR to detect the distribution of *LcGal*-*3* mRNA transcripts in 12 different tissues, with *β*-*actin* gene as an internal control. As shown in Figure 3A, *LcGal*-*3* mRNA transcripts were found to be ubiquitously distributed in various tissues, although there were some differences in expression. The highest expression was observed in the gill.

To determine the subcellular localization of the LcGal-3 protein, we constructed a eukaryotic expression vector *pEGFP*-*N1*-*LcGal*-*3* and transfected it into *Larimichthys crocea* kidney cells. We used *pEGFP*-*N1* as a negative control. The subcellular localization of the GFP-LcGal-3 protein was found in both cytoplasm and nucleus, similar to that of the EGFP control protein (Figure 3B).

### 2.3. Prokaryotic Expression and Purification of LcGal-3 Protein

To investigate the role of the LcGal-3 protein, the recombinant GST-LcGal-3 protein was expressed in *E. coli* BL21 (DE3), purified using affinity chromatography, and analyzed by SDS-PAGE, as shown in Figure 4. The GST-LcGal-3 fusion protein had a size of 60.17 kDa (with the LcGal-3 protein being approximately 34.17 kDa), while the GST tag protein had a size of 26 kDa, consistent with the predicted size.

### 2.4. Hemagglutination and Sugar Inhibition Assays of LcGal-3 Protein

Based on the results presented in Figure 5, it is evident that the purified recombinant protein effectively performs a blood coagulation test. Specifically, the rLcGal-3 protein displays a significant agglutination effect on the red blood cells of the large yellow croaker, while no significant difference in binding activity was observed with or without Ca^2+^ in the binding buffer. In contrast, the control group GST protein did not exhibit any agglutination effect on the red blood cells. Therefore, it can be concluded that the Gal-3 protein of the large yellow croaker has the ability to agglutinate red blood cells (Figure 5A), and it does not rely on Ca^2+^ for this function.

To confirm the specificity of the sugar-binding motifs H-NPR and WG-EE in the LcGal-3 protein’s function, we obtained mutated LcGal-3a and LcGal-3b, lacking H-NPR and WG-EE, respectively, through prokaryotic expression (Figure 5B). We also acquired two proteins, LcGal-3c and LcGal-3d, with point mutations in the histidine residues of H-NPR, where histidine was mutated to either proline or aspartic acid (Figure 5B). As shown in Figure 5C, LcGal-3a and LcGal-3b failed to agglutinate red blood cells, while LcGal-3c and LcGal-3d lost their agglutination activity at lower protein concentrations.

The inhibitory effect of sugar on the coagulation of LcGal-3 was used to determine the carbohydrate binding specificity of the LcGal-3 protein. The results demonstrate that the agglutination activity of the LcGal-3 protein on large yellow croaker RBCs is inhibited by lipopolysaccharide (LPS), indicating that the LcGal-3 protein can bind to LPS. The lowest inhibitory concentration of LPS is 0.0390625 μg/mL. Other sugars, such as peptidoglycan, lactose, and D-galactose, do not have any inhibitory effect on coagulation (Figure 5D). These findings highlight the specificity of LcGal-3 for carbohydrate binding.

### 2.5. Defense Response of LcGal-3 against P. plecoglossicida Infection

Fluorescent quantitative PCR technology (qRT-PCR) was used to investigate the expression level of the *LcGal*-*3* gene in the large yellow croaker infected with *P*. *plecoglossicida*. The mRNA of *LcGal*-*3* was analyzed in the liver, spleen, gills, and head-kidney tissues at different time points after infection (0 h, 12 h, 24 h, 48 h, 72 h, and 96 h). As *LcGal*-*3* is closely linked to disease resistance, its up-regulation after *P*. *plecoglossicida* infection was analyzed. The results showed that the expression of *LcGal*-*3* was significantly upregulated after infection. In particular, the expression level of *LcGal*-*3* in the liver was markedly upregulated after 48 h of infection, being 14.42 times higher than that in the control group (Figure 6). In the infected gill, spleen, and head-kidney tissues, the expression level of LcGal-3 was moderately upregulated at different time points, with 2.11 times, 4.15 times, and 3.82 times higher expression than that in the control group, respectively (Figure 6). The activated expression of LcGal-3 against P. plecoglossicida infection at the transcriptional level suggests a positive role for LcGal-3 in response to the pathogen.

### 2.6. Agglutination and Bactericidal Effect of LcGal-3 on Gram Negative Bacteria

The bacteria agglutination assay showed that the LcGal-3 protein had agglutination effects on three out of the twelve tested bacteria, which were Gram-negative bacteria commonly found in marine fish, including *P*. *plecoglossicida*, *V*. *parahaemolyticus*, and *V*. *harveyi* (as shown in Figure 7). No agglutination was observed in other bacteria such as *Staphylococcus aureus* (as shown in Figure 7), *Escherichia coli*, *Edwardsiella tarda*, *V*. *alginolyticus*, *V*. *vulnificus*, *P*. *aeruginosa*, *Aeromonas hydrophila*, *B*. *subtilis*, and *Micrococcus lysodeikticus*. These results suggest that the LcGal-3 protein has a specific bacterial agglutination activity, which could play a crucial role in microbial recognition.

To assess the viability of bacteria treated with the LcGal-3 protein, the Live/Dead^®^ BacLight^TM^ Bacterial Survivability Kit (Invitrogen, Carlsbad, CA, USA) was used. The viability of twelve bacteria incubated with LcGal-3 was evaluated using the red fluorescent dye propidium iodide (PI), which exclusively enters dead cells with a compromised membrane, while SYTO9 (green) stains all cells [19]. The viability assay clearly showed that LcGal-3 treatment resulted in the destruction of the bacterial cell wall, killing three aggregating bacteria (*P*. *plecoglossicida*, *V*. *parahaemolyticus*, and *V*. *harveyi*) (see Figure 7). In contrast, cells treated with free GST protein remained unaffected, as the bacteria did not aggregate, and the cell wall and membrane remained intact (see Figure 7). These results suggest that LcGal-3 may be an effective antibacterial agent against Gram-negative bacteria.

Furthermore, when examined using scanning electron microscopy (SEM) (as shown in Figure 8), the three aforementioned Gram-negative bacteria were observed to aggregate, with a large number of bacteria visible within the same field of view. Following treatment with the LcGal-3 protein, the bacterial cell walls were found to have shrunk and even collapsed, with the cell walls of *P*. *plecoglossicida*, *V*. *parahaemolyticus*, and *V*. *harveyi* even appearing to fuse. In contrast, the negative control group treated with GST protein and the LcGal-3a protein group lacking the CRD domain did not exhibit any aggregation (with a small number of bacteria visible within the same field of view), and their cell walls remained intact. Based on these observations, it can be concluded that the LcGal-3 protein specifically causes bacterial aggregation, ultimately leading to bacterial death by disrupting or fusing their cell walls.

## 3. Discussion

Bacterial infection, especially visceral white spot disease caused by *P*. *plecoglossicida*, has posed a significant threat to the large yellow croaker aquaculture industry. In order to decipher the genetic mechanism of antibacterial resistance in the large yellow croaker and achieve molecular breeding for antibacterial traits, we conducted a GWAS analysis and identified a set of genetic loci closely associated with disease resistance. Among the candidate genes, a unique chimeric galectin, namely LcGal-3, was identified. This study further investigated its role in the defense response against *P*. *plecoglossicida* infection. Our results demonstrated that LcGal-3 possessed agglutination activity, and this agglutination activity relied on two conserved β-galactoside-binding motifs (H-NPR and WG-EE-) in the carbohydrate recognition domain (CRD). LcGal-3 can specifically bind to LPS. Fluorescence microscopy and scanning electron microscopy (SEM) observations showed that LcGal-3 could agglutinate bacteria and exert bactericidal effects by disrupting bacterial cell membranes.

LcGal-3 is constitutively expressed in the large yellow croaker, with the highest expression levels in the gill and skin. Similar results have been observed in the Japanese pufferfish (*Takifugu rubripes*) [20]. As important mucosal immune organs, the gill and skin serve as the first line of defense against the invasion of pathogens in teleost fish [20]. Additionally, immune stimulation and inflammation can enhance the expression of galectin-3 [21,22,23]. Following stimulation by *P*. *plecoglossicida*, the expression of LcGal-3 was significantly upregulated in the liver, similar to the results observed in the yellow drum (*Nibea albiflora*) [11], Japanese pufferfish (*T*. *rubripes*) [20], and grass carp (*Ctenophyngodon idella*) [16]. These confirmed the important role of the liver in innate immune responses as a residence for various immune cells such as macrophages [24]. Furthermore, similar studies have been conducted in invertebrates. The cloned EsGal in the Chinese mitten crab (*Eriocheir sinensis*) showed relatively high expression levels in the hepatopancreas, gills, and blood cells [25]. The cloned SpGal from the mud crab (*Scylla paramamosain*) showed a rapid increase in expression after stimulation by *Vibrio alginolyticus*, reaching maximum expression levels 6 h post stimulation, with the highest expression level in the hepatopancreas [26]. It is thus speculated that Gal-3 plays important roles in the host immune responses in vertebrates as well as in invertebrates, although the exact function may vary among different species.

The agglutination assay is a functional method used to evaluate the activity of lactose-binding lectins. The LcGal-3 protein can agglutinate red blood cells of the large yellow croaker, and this agglutination activity is inhibited by LPS, indicating that LcGal-3 can specifically bind to LPS. LPS is the main component of the cell wall of Gram-negative bacteria, and consists of a core polysaccharide, O-polysaccharide side chains, and lipid A [27]. There are differences in LPS composition among different groups and even strains of bacteria. It is speculated that the bactericidal mechanism of LcGal-3 is similar to that of lysozyme, where the cations in LcGal-3 molecules can interact with the divalent cations binding to LPS in the bacterial cell wall. Since the affinity of the cations in LcGal-3 molecules for LPS is higher than that of Ca^2+^ and Mg^2+^ in the cell wall, binding to LPS, the abundant cations carried by LPS can be competitively replaced by lysozyme, leading to the destruction of the cell membrane barrier and the subsequent release of various molecules, such as small proteins and hydrophobic compounds, especially antibiotics, into the bacterial cell, resulting in bacterial death [28,29].

LcGal-3 from the large yellow croaker exhibits agglutination and bactericidal effects against its main pathogens, *P*. *plecoglossicida*, *V*. *parhemolyticus*, and *V*. *harveyi*, which are Gram-negative bacteria. Similar results have been obtained in yellow drum (*N*. *albiflora*) [11], Japanese pufferfish (*T*. *rubripes*) [20], and grass carp (*T*. *idella*) [15]. In addition, rEsGal from the Chinese mitten crab (*E*. *sinensis*) can bind to LPS, PGN, and GLU even at relatively low concentrations, and it exhibits a strong agglutination activity against two Gram-negative bacteria, two Gram-positive bacteria, and a fungi (*Pichia pastoris*) [25]. SpGal cloned from the mud crab (*S*. *paramamosain*) shows a dose-dependent high affinity for LPS and also exhibits agglutination activity against three Gram-negative bacteria and three Gram-positive bacteria [26]. The CgGal-3 protein from the Pacific oyster (*Crassostrea gigas*) has a high affinity for various PAMPs such as glucan, LPS, and PGN, and it can bind to two Gram-negative bacteria, two fungi (*Saccharomyces cerevisiae* and *Pichia pastoris*), and a Gram-positive bacterium (*Micrococcus luteus*) [30]. Galectin-3 binds to exogenous polysaccharides on the surfaces of potential pathogenic microorganisms, parasites, and fungi, and acts as a pattern recognition receptor (PRR) to exert immune functions [31,32]. In the innate immune response, pattern recognition receptors (PRRs) are the primary means by which innate immune responses identify damage or pathogen-associated molecular patterns (PAMPs), such as fungal and bacterial PGN, LPS, viral genomes, and damaged vesicles [33,34]. Pathogenic microorganisms in nature have different types of sugar substances on their cell surfaces, including LPS and PGN. Galectins, which are galactoside-binding proteins with specific recognition, binding, and agglutination functions for polysaccharides, play an important role in the body’s defense against foreign microorganisms [35]. Considering the agglutination and antimicrobial properties of lectins found in various aquatic organisms, this provides new ideas for developing them into immune enhancers and antimicrobial agents.

Further research has revealed that Galectin-3 can bind to β-1,2-linked oligomannosides present on the surface of *Candida albicans*, resulting in direct fungal death, which is evident from cell shrinkage and damage to membrane integrity [36]. Galectin-3 also exhibits a direct lytic effect on *Cryptococcus neoformans*, a fungus that causes cryptococcosis, and inhibits its growth [37]. In gastric epithelial cells infected with Helicobacter pylori, the secretion of Galectin-3 by the cells can bind to the bacteria’s LPS, prompting phagocytes to aggregate rapidly at the site of infection [38]. Furthermore, Galectin-3 can impede the growth of *Paracoccidioides brasiliensis* and diminish the stability of its extracellular vesicles, thereby playing a crucial role in defending the host against pathogen infections [39]. The antibacterial mechanism of LcGal-3 in the large yellow croaker against its main pathogen *P*. *plecoglossicida* and other bacteria needs further investigation. It may affect the integrity and permeability of the cell wall and plasma membrane [40,41], triggering the production of reactive oxygen species (ROS). As a response to potential DNA damage caused by oxidative stress and excessive ROS, the expression of RecA is upregulated. The increase in RecA levels stimulates the autocatalytic cleavage of the transcriptional repressor LexA bound to ssDNA, activating the SOS response in bacterial cells [42]. Under severe oxidative stress, the accumulation of free radicals can lead to DNA double-strand breaks and depolarization of the cell membrane, ultimately resulting in bacterial cell death [43]. Fish are frequently exposed to water containing diverse pathogens, and Galectin-3 such as LcGal-3 can be secreted onto the mucosal surface via non-classical pathways, participating in the immune response and exerting antibacterial functions. These findings suggest that Galectin-3 plays a vital role in the immune system of fish, particularly in innate immunity.

## 4. Materials and Methods

### 4.1. Experimental Fish and Bacterial Challenge Experiment

Healthy large yellow croakers, weighing 270 ± 9.0 g, were raised in a seawater farm located in Ningde City, Fujian Province, China. Twelve different tissues, including gills, skin, brain, swim bladder, heart, intestines, spleen, head-kidney, body kidney, liver, stomach, and muscle, were collected from three fish and stored at −80 °C for further analysis.

For the *P*. *plecoglossicida* infection experiment, the large yellow croakers were acclimatized for 10 days in a 4 m^3^ water tank with a temperature range of 23–26 °C and a salinity range of 25–26 PSU. 

After inoculation of large yellow croakers with *P*. *plecoglossicida* bacteria at a final concentration of 1.0 × 10^6^ cfu/mL, liver, spleen, gill, and head-kidney tissues were collected at different time points: 0 h, 12 h, 24 h, 48 h, 72 h, and after immune stimulation. The collected tissues were immediately frozen using liquid nitrogen, and stored at −80 °C. Kidney and brain tissues collected after 96 h of pathogenic attack were placed in 4% paraformaldehyde at room temperature for immunohistochemistry. Samples collected from unchallenged fish (healthy fish) were included as control.

### 4.2. RNA Extraction and cDNA Synthesis

Total RNA was isolated from the above samples using *TransZol* Up Plus RNA Kit (TransGen Biotech, Beijing, China), according to the manufacturer’s instruction. First-strand cDNA was synthesized from total RNAs by GoScript™ Reverse Transcription System (Promega, Madison, WI, USA) in accordance with the manufacturer’s protocols.

### 4.3. Primer Design and Clone of LcGal-3 Gene

The open reading frame (ORF) of *LcGal*-*3* was screened from the transcriptome database of *L*. *crocea*, which was constructed in our laboratory. To clone the ORF, we designed a pair of specific primers (Gal-3-F and Gal-3-R, Table 2) with *EcoR* I and *Xho I* restriction sites, following the protocol of the ClonExpress^®^ II One Step Cloning Kit (Vazyme Biotech, Nanjing, China). PCR was performed as follows: an initial denaturation step at 95 °C for 3 min, followed by 30 cycles of 95 °C for 30 s, 55 °C for 30 s, and 72 °C for 1 min 45 s, and a final extension step at 72 °C for 5 min. The *LcGal*-*3* ORF was then purified from the gel and recombined into the *pGEX*-*6P*-*1* vector, which was cut by the same restriction sites. The resulting construct was sent to BioSune (Shanghai, China) for sequencing.

### 4.4. Bioinformatics Analysis

The Gal3 protein sequences were retrieved from the NCBI website (https://www.ncbi.nlm.nih.gov, accessed on 2 April 2021). Clustal Omega (https://www.ebi.ac.uk/Tools/msa/clustalo/, accessed on 14 July 2021) was applied for the analysis of homology.

ExPASy-ProtParam (https://web.expasy.org/protparam/, accessed on 5 June 2021) was used to analyze the physical and chemical properties of the protein, including molecular weight, theoretical pI, and amino acid composition. SMART (http://smart.embl-heidelberg.de/smart/set_mode.cgi?NORMAL=1, accessed on 22 June 2021) was employed for the analysis of the conserved domains. SWISS-MODEL (http://swissmodel.expasy.org/interactive, accessed on 10 June 2021) was used to predict the tertiary structures. Signal peptides were predicted using the SignaIP program (http://www.cbs.dtu.dk/services/SignalP/, accessed on 7 June 2021). A phylogenetic tree was constructed using the maximum likelihood method at MEGA 6.06. Select the Gal3 protein sequence of representative species of bony fish, amphibians, birds and mammals, and construct the phylogenetic tree. Construct the phylogenetic tree using MEGA-7 software, with specific parameters including Neighbor Joining (N-J) and Bootstrap for 1000 repeated tests.

### 4.5. Tissue Distribution of LcGal-3 and Temporal Expression Pattern Post Infection

To determine the tissue-specific expression of *LcGal*-*3* in healthy fish and its expression profiles after *L*. *crocea* challenge, we performed quantitative reverse transcription PCR (qRT-PCR). We used a specific primer pair (qGal-3-F and qGal-3-R, Table 2) to amplify the *LcGal*-*3* fragment, and selected *β*-*actin* as the reference gene (β-actin-F and β-actin-R, Table 2). qRT-PCR was performed using ChamQTM Universal SYBR^®^ qPCR Master Mix (Vazyme Biotech, China) on an Applied Biosystems QuantStudio 6&7 Real-time PCR System (Application Biosystems, USA). The qRT-PCR conditions were as follows: pre-denaturation at 95 °C for 30 s, followed by 40 cycles of 95 °C for 10 s and annealing at 60 °C for 30 s. All reactions were run in triplicate, and the relative quantities were quantified on a relative scale using the 2^−ΔΔct^ method.

### 4.6. Cell Culture and Subcellular Localization

*Larimichthys crocea* kidney cells were cultured in L-15 basal medium (Leibovitzs, Thermo Fisher Scientific, Waltham, MA, USA), supplemented with 10% fetal bovine serum and 1% penicillin (100 units/mL) and 100 ug/mL streptomycin. To investigate the subcellular localization of LcGal-3, the recombinant plasmid *pEGFP*-*N1*-*LcGal*-*3* (sGal-3-F and sGal-3-R, Table 2) was introduced into *Larimichthys crocea* kidney cells, using standard procedures [19]. Specifically, the cells were plated in 12-well plates with sterile coverslips and incubated at 28 °C for 24 h. When the cell density reached 80%, the recombinant plasmid and the control plasmid pEGFP-N1 were added to the 12-well plates in proportion, using Lipofectamine^3000^ transfection reagent (Invitrogen, Carlsbad, CA, USA). The cells were then incubated for an additional 24 h under the same conditions.

To prepare for imaging, the cells were washed twice with 1 × PBS buffer (pH 7.4), fixed with 4% paraformaldehyde for 10 min at room temperature, permeabilized for 15 min, stained with DAPI (1 µg/mL) for 10 min, and washed three times with 1 × PBS buffer. Finally, the cells were observed and photographed using a fluorescence microscope (Leica SP8, Wetzlar, Germany).

### 4.7. Prokaryotic Expression and Purification of LcGal-3 Protein and Deletion Mutant LcGal-3a Protein

Recombinant plasmid *pGEX*-*6P*-*1*-*LcGal*-*3* and its deletion mutant *pGEX*-*6P*-*1*-*LcGal*-*3a*, along with the empty vector control *pGEX*-*6P*-*1*, were transformed into *Escherichia coli* BL21 (DE3) for prokaryotic expression. When the OD_600_ value of the expanded culture reached 0.6–0.8, isopropyl β-D-thiogalactoside (IPTG) was added to a final concentration of 0.01 mM, and the cultures were induced at 20 °C with shaking at 200 rpm, for 16 h.

The bacteria were then harvested by centrifugation at 7000 g for 5 min at 4 °C, washed three times with phosphate-buffered saline, and resuspended in 20 mL of phosphate-buffered saline. The cells were then lysed on ice using ultrasonication (50% power, 30 min), and the supernatant was collected by centrifugation at 12,000× *g* for 10 min at 4 °C. Glutathione Sepharose 4B (GE Healthcare) was added to the supernatant for incubation, followed by washing with equilibrium buffer. GST protein expressed by *pGEX*-*6P*-*1* was used as a negative control.

Protein expression and purification were analyzed by SDS-PAGE (12%). The protein samples were dialyzed, freeze dried, and stored at −80 °C.

### 4.8. Hemagglutination and Glucose Inhibition Assays of LcGal-3 Protein and Deletion Mutant Protein LcGal-3a

The blood coagulation process of the LcGal-3 protein was performed using a previously reported method [19]. Large yellow croaker blood was collected, and 1 mg/mL of sodium heparin solution was added as an anticoagulant. The red blood cells were washed three times with TBS buffer (20 mM Tris, 150 mM NaCl, pH 8.0) and TBS with Ca^2+^ buffer (10 mM CaCl_2_, pH 8.0) respectively, using centrifugation at 1000× *g* for 5 min. The upper layer of red blood cells was aspirated, and a cell suspension was prepared by adjusting the cells to 2% in TBS buffer. Next, 50 μL of the red blood cell suspension was mixed with the cell suspension in a 96-well U-shaped plate. Negative controls consisting of GST protein in TBS buffer and TBS with Ca^2+^ buffer were included. The mixture was incubated at room temperature for 2 h, and the blood coagulation of the red blood cells was observed using the TCS SP8 system (Leica, Germany). The experiment was conducted three times, for accuracy.

To confirm the specificity of the sugar-binding motifs (H-NPR and WG-EE-) in the LcGal-3 protein, we generated point mutations and deletion mutants. Specifically, we created two mutants: LcGal-3a, which lacks the H-NPR motif in the N-CRD, and LcGal-3b, which lacks the WG-EE- motif in the N-CRD. In addition, we produced two point mutations named LcGal-3c (with histidine in H-NPR in N-CRD mutated to proline) and LcGal-3d (with histidine in H-NPR in N-CRD mutated to aspartic acid).

To evaluate the effect of each mutant on blood clotting of red blood cells, we used the same method. Next, we tested the sugar suppression activity of the mutants, based on the clotting assay. We used four types of carbohydrate derivatives (peptidoglycan, lipopolysaccharide, lactose, and D-galactose) as substrates. We diluted each carbohydrate and mixed it with LcGal-3 protein, incubating the mixture for 1 h at room temperature. After incubation, we observed the inhibition of blood clotting caused by the carbohydrates.

### 4.9. Bacterial Agglutination and Antibacterial Assay

The purified LcGal-3 protein was tested for bacterial agglutinating and antibacterial activity. Bacteria were cultured to a late log phase, collected by centrifugation at 3000× *g* for 10 min, washed three times, and re-suspended in 0.85% NaCl buffer at a concentration of 2 × 108 cfu/mL. The bacterial suspension was mixed with LcGal-3 protein and incubated at room temperature for 2 h. The bacterial viability was assessed using the Live/Dead® BacLight™ Bacterial Viability Kit (Invitrogen). Specifically, the bacterial suspension was mixed with SYTO 9 stain (6 µM) and propidium iodide (30 µM) and incubated in the dark for 15 min. The stained bacteria were imaged using the Leica TCS SP8 system (Leica, Germany).

To examine the effect of LcGal-3 protein on bacteria, the protein (50 μL) was mixed with bacteria and incubated at room temperature for 2 h on polystyrene slides. After incubation, the samples were fixed with 2.5% glutaraldehyde for 3 h, dehydrated in a series of ethanol solutions (50%, 70%, 90%, 95%, and 100%), dried overnight, and coated with Au-Pd. The coated samples were then observed using a scanning electron microscope (SEM, FEI, Verios G4).

### 4.10. Statistical Analysis

Statistical analysis was conducted using IBM SPSS Statistics version 20.0 (IBM Corp., Armonk, NY, USA). All experiments were repeated three times. One-way ANOVA and LSD Multiple Comparison Test were utilized to assess significant differences between the challenged groups and the control groups at a 5% significance level.

## Figures and Tables

**Figure 1 ijms-24-11539-f001:**
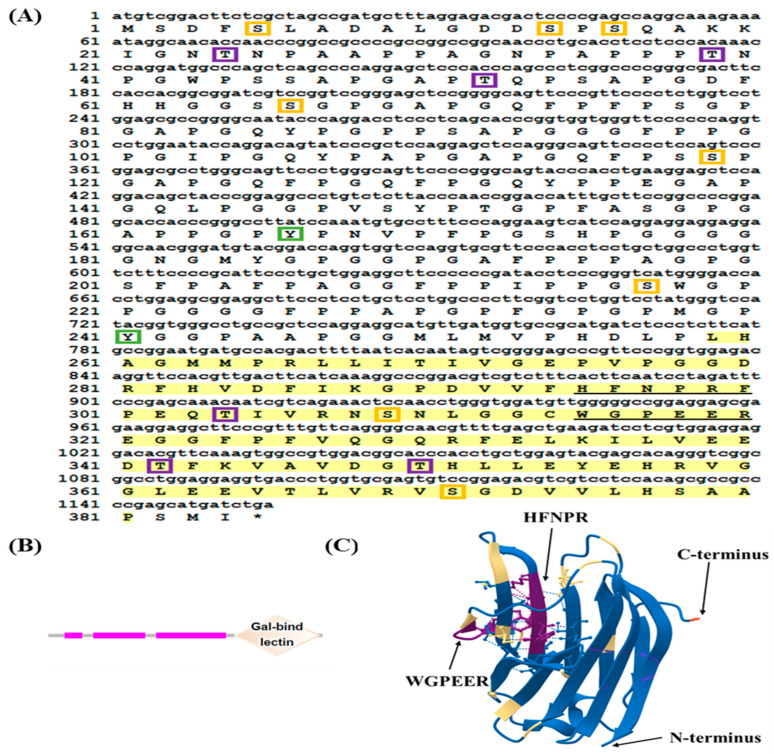
Nucleotide sequence and deduced amino acid sequence and the structure of LcGal-3. (**A**) The stop codons are marked with an asterisk (*), and the CRDs are highlighted in yellow. The characteristic β-galactoside binding motifs, H-NPR and WG-EE-, are underlined. Serine phosphorylation sites are indicated by orange boxes, threonine phosphorylation sites by purple boxes, and tyrosine phosphorylation sites by green boxes. (**B**) The N-CRD and C-CRD protein domains of LcGal-3 were predicted using the SMART database. The three pink bands represent the low complexity region and the rhombus represents the carbohydrate recognition (CRD) domain. (**C**) The tertiary structure of the LcGal-3 protein, including the sugar binding motifs H-NPR and WG-EE-, as well as the C-terminus and N-terminus are shown. Random coils and β-turns are highlighted in blue and yellow, respectively.

**Figure 2 ijms-24-11539-f002:**
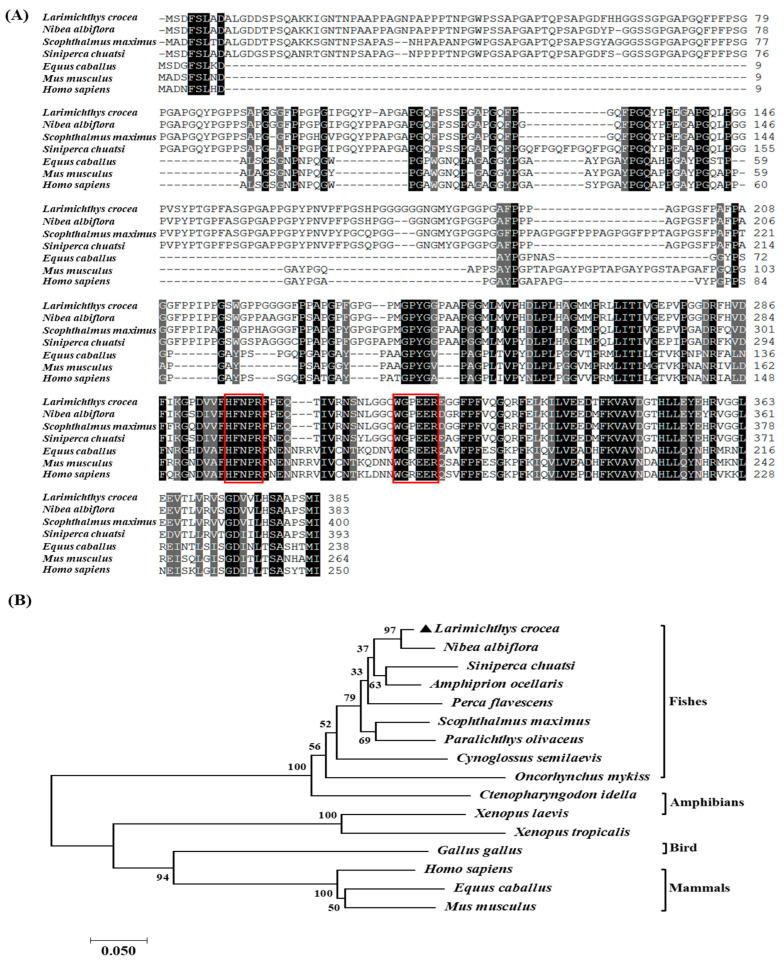
Sequence alignment and phylogenetic analysis of LcGal-3 and its homologs in other species. (**A**) Multiple sequence alignment of LcGal-3 and its homologs. The red boxes indicate the sugar binding motifs (H-NPR and WG-EE-), the black shadows indicate an identical amino acid, and the GenBank accession numbers of the amino acid sequences of the species are listed in Table 1. (**B**) Phylogenetic analysis of LcGal-3 and its homologs. The numbers at the nodes represent the bootstrap confidence values (100%) of 1000 replicates. The galectins are included as an outgroup. The scale bar (0.050) indicates the genetic distance. The GenBank accession numbers of amino acid sequences used are listed in Table 1.

**Figure 3 ijms-24-11539-f003:**
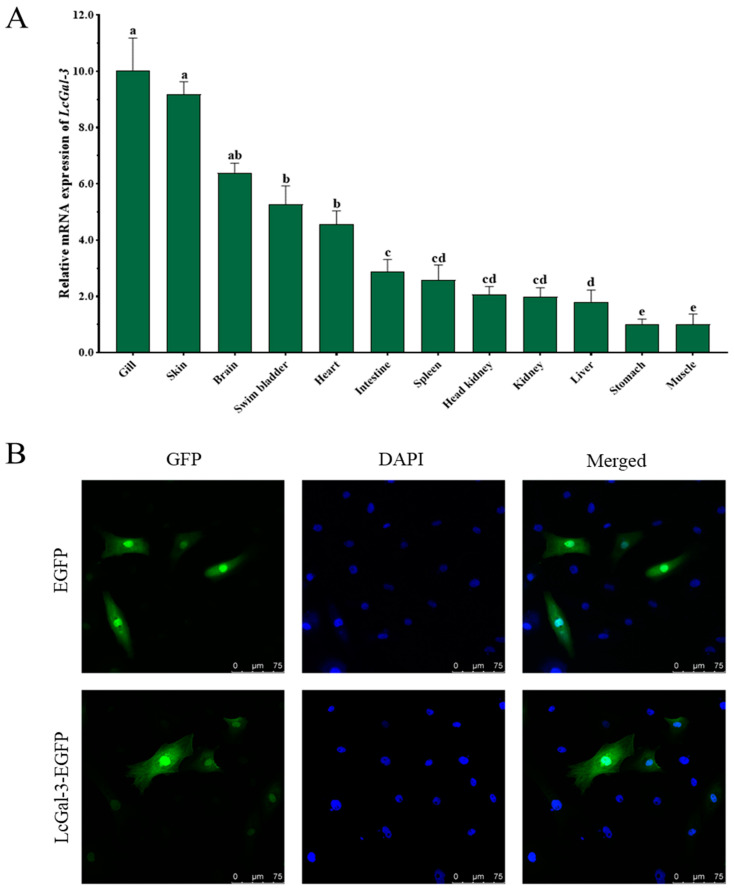
Tissue specificity and subcellular localization of LcGal-3. (**A**) Tissue expression pattern of *LcGal*-*3* in large yellow croaker. *β*-*actin* was used as an internal control. Values represent mean ± SD (standard deviation) of three biological replicates. The letters a, b, c and d denote statistical significance (*p* < 0.05). (**B**) Subcellular localization of LcGal-3 in transfected *Larimichthys crocea* kidney cells. Images were captured by laser confocal microscopy.

**Figure 4 ijms-24-11539-f004:**
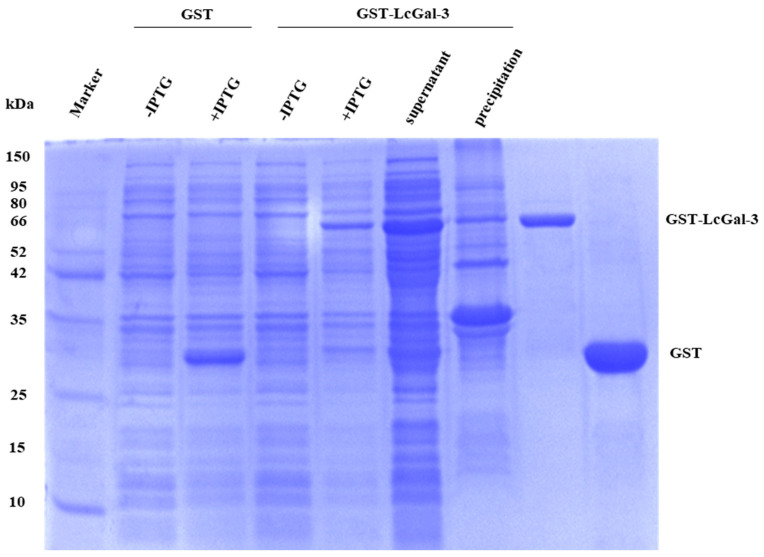
SDS-PAGE analysis of the recombinant LcGal-3 expression in *E. coli* BL21.

**Figure 5 ijms-24-11539-f005:**
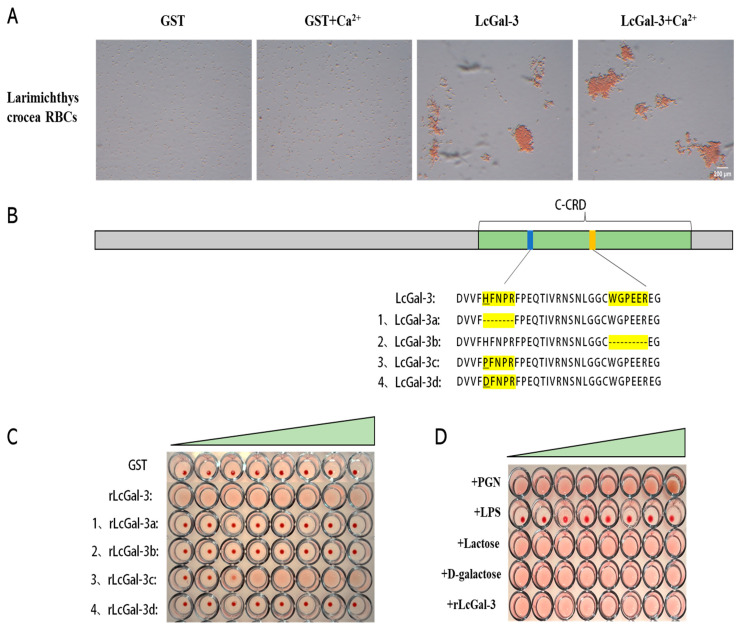
Hemagglutination and sugar inhibition assays of LcGal-3 protein and its mutant proteins. (**A**) Hemagglutination of LcGal-3 protein toward RBCs from *L*. *crocea* (large yellow croaker). The GST protein groups were used as negative controls. (**B**) Schematic of sugar binding motifs rLcGal-3 deletion mutants rLcGal-3a and rLcGal-3b, site-directed mutants rLcGal-3c and rLcGal-3d. (**C**) Hemagglutination assay of mutant rLcGal-3 proteins towards RBCs of *L*. *crocea*. rLcGal-3,rLcGal-3b and rLcGal-3d cannot agglutinate RBCs; rLcGal-3c can agglutinate RBCs partially. The triangle represents the concentration of the recombinant protein, diluted in a double gradient from right to left; the added proteins are 500 μg/mL, 250 μg/mL, 125 μg/mL, 62.5 μg/mL, 31.25 μg/mL, 15.625 μg/mL, 7.8125 μg/mL, and 3.90625 μg/mL, respectively. (**D**) Sugar inhibition assay of rLcGal-3 protein; only LPS has an inhibitory effect on agglutination activity of rLcGal-3 protein. The triangle represents the concentration of the recombinant protein, diluted in a double gradient from right to left; the added proteins are 500 μg/mL, 250 μg/mL, 125 μg/mL, 62.5 μg/mL, 31.25 μg/mL, 15.625 μg/mL, 7.8125 μg/mL, and 3.90625 μg/mL, respectively.

**Figure 6 ijms-24-11539-f006:**
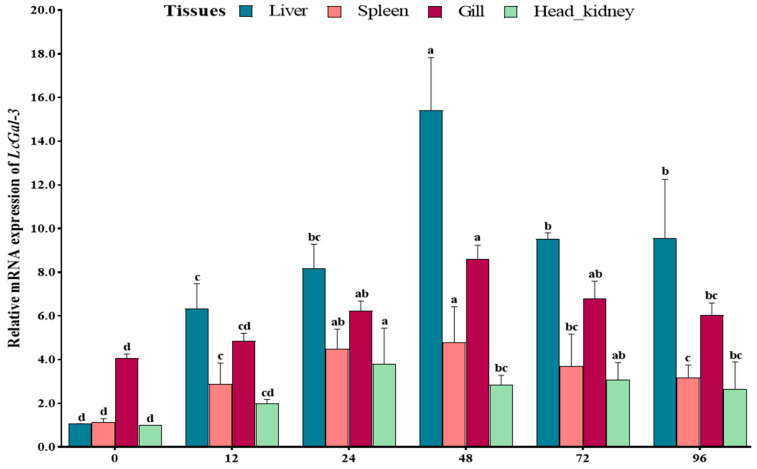
Defense response of LcGal-3 against *P. plecoglossicida* infection. Time-course expression of LcGal-3 in different tissues after *P. plecoglossicida* infection: the relative expression levels of LcGal-3 were determined by qRT-PCR in the spleen, head-kidney, brain, and liver at 0, 12, 24, 48, 72, and 96 h after *P. plecoglossicida* challenge. *β-actin* was used as an internal control. The data are presented as mean ± SD (n = 6). Statistical significance is represented by the letters a, b, c and d (*p* < 0.05).

**Figure 7 ijms-24-11539-f007:**
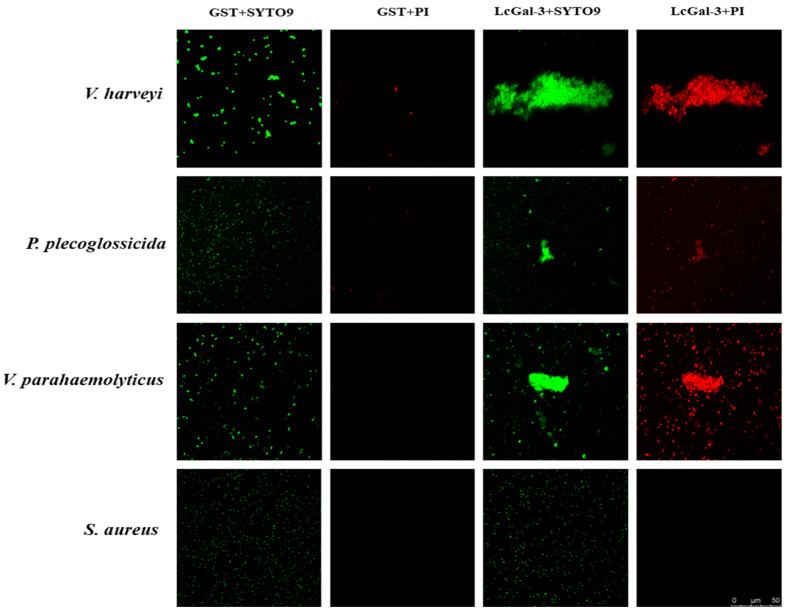
Bacterial agglutination and cell viability staining by LcGal-3 protein. All the bacteria stained by SYTO 9 are in green, and dead bacteria stained by propidium iodide (pI) are in red. Images were captured under a fluorescent microscope. Non-agglutinated bacteria representative: *S*. *aureus*.

**Figure 8 ijms-24-11539-f008:**
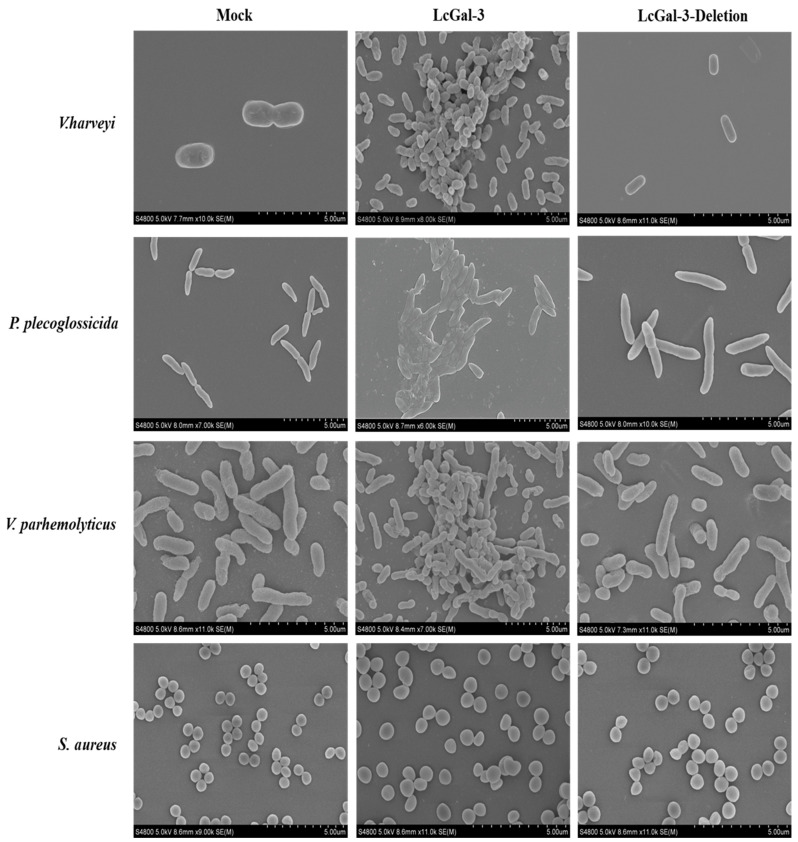
SEM images of antibacterial activity of rLcGal-3 and mutant rLcGal-3 proteins interacted with bacteria. The bacteria include *V*. *harveyi*, *P*. *plecoglossicida*, *V*. *parahaemolyticus* and *S*. *aureus* in control group (**left**), treatment with rLcGal-3 protein (**middle**) and treatment with deletion mutant rLcGal-3 (**right**). Non-agglutinated bacteria representative: *S*. *aureus*.

**Table 1 ijms-24-11539-t001:** Identity analysis of amino acid sequences of Galectin-3 protein in *Larimichthys crocea* and other species.

Species	Common Name	GenBank Accession number	Identity (%)
*Larimichthys crocea*	Large yellow croaker	XP_027130834.1	100
*Nibea albiflora*	Yellow drum	MW858251.1	95.56
*Perca flavescens*	Yellow perch	XP_028421130.1	91.85
*Scophthalmus maximus*	Breet	MF797865.1	91.85
*Paralichthys olivaceus*	Bastard Halibut	XP_019962524.1	91.85
*Amphiprion ocellaris*	Clownfish	XP_023142436.1	90.30
*Siniperca chuatsi*	Aucha perch	XM_044167408.1	87.41
*Cynoglossus semilaevis*	Tongue sole	XP_016891389.1	84.33
*Ctenopharyngodon idella*	Grass carp	XP_051772445.1	77.04
*Oncorhynchus mykiss*	Rainbow trout	XP_036818451.1	73.76
*Mus musculus*	Mouse	NM_010705.3	50.36
*Gallus gallus*	Chicken	XM_046917600.1	50.00
*Equus caballus*	Horse	XM_023627775.1	49.64
*Xenopus laevis*	Clawed frog	BC046662.1	47.83
*Xenopus tropicalis*	Xenopus tropicalis	NM_203655.1	46.72
*Homo sapiens*	Human	BAB83625.1	43.32

**Table 2 ijms-24-11539-t002:** Primers used in this study.

Primer Name	Sequence (5′-3′)	Purpose
Gal-3-F	CCCCTGGGATCCCCGGAATTCATGTCGGACTTCTCGCTGGC	ORF amplification
Gal-3-R	GTCACGATGCGGCCGCTCGAGTCAGATCATGCTCGGGGCG	
qGal-3-F	AGGAGTCCATCTTCTGATCATGT	qRT-PCR analysis
qGal-3-R	ATCCTGGGTTTGTGGGAGGA	
β-actin-F	TTATGAAGGCTATGCCCTGCC	
β-actin-R	TGAAGGAGTAGCCACGCTCTGT	
sGal-3-F	CTACCGGACTCAGATCTCGAGATGTCGGACTTCTCGCTGGC	subcellular localization
sGal-3-R	GTACCGTCGACTGCAGAATTCGGATCATGCTCGGGGCGGC	
HFNPR-F	AGAGCCGACGTGGCGTTTTTCAAACGCT	deletion mutations
HFNPR-R	AGCGTTTGAAAAACGCCACGTCGGCTCT	
PFNPR-F	TTTCCCTTCAACCCGAGGTTCAAACGCTCG	point mutations
PFNPR-R	CGAGCGTTTGAACCTCGGGTTGAAGGGAAA	
DFNPR-F	GCGTTTGACTTCAACCCGAGGTTCAAACGCT	
DFNPR-R	AGCGTTTGAACCTCGGGTTGAAGTCAAACGC	

Note: EcoR I restriction site (GAATTC) and Xho I enzyme restriction site (CTCGAG) are underlined.

## Data Availability

Not applicable.

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
