# Peer review of "Molecular Characterization of Galectin-3 in Large Yellow Croaker Larimichthys crocea Functioning in Antibacterial Activity"

_ijms, 2023, doi:10.3390/ijms241411539_

Round 1
Reviewer 1 Report
To the authors, I congratulate the elaboration and techniques used for the presented draft. The draft has technical competence and brings relevant results in immunity for the studied species (Larimichthys crocea), which is of economic importance for the country (China).
My considerations will be more emphasized in MM and results.
I believe that the draft needs to have more careful attention when presenting the data, and when bringing this in written form, for example, the phylogeny of a gene that is its first time isolated (LcGal-3), with only 5 possibilities for comparison between fish, is it not representative, and the possible isoforms with which they could be associated? to have greater veracity, it must have greater possibilities in phylogeny, with greater clades represented.
in fact I am not an expert in expression and purification of protein, but I believe that the image in figure 4 has a very large drag, possibly they could have a cleaner image (I say because if it were PCR, this drag would be non-specificity of the primer, which should be redesigned).
In the writing
In introduction, i believe that the beginning of the first paragraph is not very inviting, I believe it should be written.
When describing the MM of phylogeny and structure prediction, basically they just sit the program used without any details of comparisons used, in the case of phylogeny, they did not give details of which animals were compared, if the gaps were disregarded in the alignment, resulting in a tree with little impact as a result.
Discussion is well presented with linearity and impactful to read!!
Summarizing, the draft has technical competence and brings relevant results for the studied species, which is of economic importance for the country, but needs small changes for better quality;
I ask you to consider the small attentions in the images and remake the phylogeny with greater possibilities of association, which will describe reality with greater capacity. The writing of the introduction can be more appealing to the reader as well as greater detail of the MM.
Reviewer 2 Report
The authors identified galectin-3 gene of yellow croaker and studied the antibacterial activity. I think that the experiments were carried out systemically and in the right method. However, the discussion part was very worthless because it consists of only a series of references (findings in other fish). For example, the authors should mention how the results can contribute to aquaculture of this fish, what characteristics does Gal-3 of this fish have and so on. The discussion part must be rewritten from the beginning.
<Introduction>
L42
Please explain ‘chimeric galectin’ in detail.
L45-68
Characterization of galectin-3 have been performed in other fish. Why is Gal-3 of yellow croaker needed to be characterized?
L81-83
Please discuss about ‘new insights’ in discussion part.
<Results>
Figure 1B
I could not find a hexagon.
Figure 1 legend
Please write the title of figure before the explanations.
L132
The highest expression is the gill in Figure 3.
L133-142
HEK293T cells are derived from human embryonic kidney. I could not understand why the authors used these cells to know intercellular localization of LcGal-3. If the authors would like to know the localization, the authors should perform immunohistochemistry using the tissues of yellow croaker. Thus, I think that this experiment is meaningless. Indeed, the authors have the antibody as they used it in Fig. 6.
Figure 4
Please write treatment of rightest two lanes.
Figure 5
What does the triangle denote?
Figure 5
Please explain how I should observe the results? I see that the red substances are aggregation.
L207
Why did the authors not perform this study using the liver? LcGal-3 mRNA expression was highest in the liver, and it was not using the kidney and the brain.
Whole
Please use uniformly SD or SE in Figures
<Discussion>
As mentioned in the first, please revise the whole. The authors’ description is not ‘discussion’. For example, as the 2nd paragraph (line270-) consist of only general mention (-line271), your data (-line274) and other fish data (-288), there is no discussion. This is true of all paragraph.
I could read the manuscript smoothly. However, as I am not a native speaker, minor errors may be checked by a native speaker.
